# *Dishevelled* Has Anti-Viral Activity in Rift Valley Fever Virus Infected *Aedes aegypti*

**DOI:** 10.3390/v15112140

**Published:** 2023-10-24

**Authors:** Christian B. Smith, Natasha F. Hodges, Rebekah C. Kading, Corey L. Campbell

**Affiliations:** Center for Vector-Borne Infectious Diseases, Department of Microbiology, Immunology and Pathology, College of Veterinary Medicine and Biomedical Sciences, Colorado State University, Fort Collins, CO 80523, USA; christian.smith@colostate.edu (C.B.S.); rebekah.kading@colostate.edu (R.C.K.)

**Keywords:** vector biology, anti-viral immunity, cell signaling, vector competence, transcriptional regulation

## Abstract

Mosquitoes in the genera *Aedes* and *Culex* are vectors of Rift Valley fever virus (RVFV), which emerges in periodic epidemics in Africa and Saudi Arabia. Factors that influence the transmission dynamics of RVFV are not well characterized. To address this, we interrogated mosquito host-signaling responses through analysis of differentially expressed genes (DEGs) in two mosquito species with marked differences in RVFV vector competence: *Aedes aegypti* (*Aae*, low competence) and *Culex tarsalis* (*Cxt*, high competence). Mosquito–host transcripts related to three different signaling pathways were investigated. Selected genes from the Wingless (Wg, WNT-beta-catenin) pathway, which is a conserved regulator of cell proliferation and differentiation, were assessed. One of these, *dishevelled* (*DSH*), differentially regulates progression/inhibition of the WNT and JNK (c-Jun N-terminal Kinase) pathways. A negative regulator of the JNK-signaling pathway, puckered, was also assessed. Lastly, Janus kinase/signal transducers and activators of transcription (JAK-STAT) are important for innate immunity; in this context, we tested domeless levels. Here, individual *Aae* and *Cxt* were exposed to RVFV MP-12 via oral bloodmeals and held for 14 days. Robust decreases in DEGs in both *Aae* and *Cxt* were observed. In particular, *Aae DSH* expression, but not *Cxt DSH*, was correlated to the presence/absence of viral RNA at 14 days post-challenge (dpc). Moreover, there was an inverse relationship between the viral copy number and *aaeDSH* expression. *DSH* silencing resulted in increased viral copy numbers compared to controls at 3 dpc, consistent with a role for *aaeDSH* in antiviral immunity. Analysis of cis-regulatory regions for the genes of interest revealed clues to upstream regulation of these pathways.

## 1. Introduction

*Aedes aegypti aegypti* (*Aae*) and *Culex tarsalis* (*Cxt*) are competent vectors of Rift Valley fever virus (RVFV, *Phenuiviridae*: *Phlebovirus*), a zoonotic virus that periodically causes epidemics in Africa and Saudi Arabia [1,2,3]. RVFV infects primarily ruminants [4], however, spillover into humans also occurs. Most human infections result in mild illness [5], however, kidney and liver damage, ocular pathology, severe anemia, hemorrhagic fever and miscarriage can occur [6,7]. RVFV epizootic outbreaks have so far been confined to Africa and the Arabian peninsula [6,8,9], however, global travel and climate change could allow emergence and establishment into new regions [10,11,12].

Dozens of mosquito species have been associated with RVFV infection in the field, with many species capable of contributing to transmission during outbreaks [13]. For example, about 20, 8 and 12 *Aedes*, *Anopheles* and *Culex* species, respectively, have been implicated, based on the presence of viral genomes in field-collected specimens (reviewed in [14]). Other field collections included *Eretmapodites* (*n* = 4) and *Mansonia* (*n* = 2) species. Nevertheless, mosquito populations show a wide range of susceptibility levels in the laboratory. Our previous work, which compared RVFV dynamics between *Aae* Poza Rica and the *Cxt* Kern National Wildlife Refuge (KNWR) strain, showed markedly different levels of viral RNA (vRNA) and infectious virus at 14 days post-exposure to artificial bloodmeals containing approximately equal titers of RVFV MP-12 [15]. Whereas 100% of *Cxt* were positive for vRNA, just 62% of *Aae* carcasses were vRNA positive. Similarly, though 100% of *Cxt* had detectable vRNA in their legs/wings, demonstrative of dissemination, just 33% of *Aae* were vRNA positive. Lastly, when tested for transmission potential, 47% of *Cxt* individuals had infectious virus in saliva expectorants, whereas *Aae* had a 16% positivity [15]. Therefore, we hypothesized that RVFV-responsive host cell-signaling processes may differ between mosquito species and thus contribute to the observed phenotypic variability in vector competence.

An RNA interference (RNAi) screen of human cell culture had previously reported that WNT (Wingless (Wg) and *INT1*) signaling was increased during RVFV infection [16]. The WNT/beta-catenin pathway is an evolutionarily conserved pathway that controls cellular proliferation, development and cell self-renewal across taxa. Several studies have also demonstrated that viruses may manipulate this pathway to promote infection [16,17,18]. In humans, a RVFV protein transcriptionally activates beta-catenin [16]. Most published examples from studies of human cells indicate modulation of the WNT pathway during infection with a wide variety of viruses, including human immune-deficiency virus, influenza A as well as hepatitis A, B and C [17,18,19,20]. We chose to explore the role of this and other signaling pathways during mosquito RVFV infection. For this work, we tested archival samples collected in our previous characterization of RVFV MP-12 viral loads [15]. Here, we focused on differential expression of transcripts coding for effectors of mosquito–host signaling pathways. Our genes of interest (GOIs) included armadillo (*ARM*, ortholog of beta-catenin), frizzled2 (*FZ2*) and dishevelled (*DSH*) of the WNT cell-signaling pathway, domeless (*DOME*) of the JAK-STAT (Janus kinase/signal transducers and activators of transcription) pathway and puckered (*PCK*) of the jun-kinase (JNK) pathway (Figure 1, accession numbers listed in Appendix A). In mosquitoes, the JNK pathway has been implicated in restriction of arbovirus replication in flavivirus and alphavirus-infected mosquitoes [21].

*ARM*, *FZ2* and *DSH* are all positive regulators of the WNT signaling pathway. Thus, differential expression of each of these transcripts may lead to enhanced cell signaling that could alter infection parameters. *DSH* differentially regulates progression of either the JNK or WNT pathways through interactions with the negative regulator naked cuticle [22]. *DOME* is a positive regulator of the JAK-STAT pathway and is important for the localization of STAT dimer proteins at receptor proteins in the cell membrane, which are then transported to the nucleus for activation of downstream pathway targets. *PCK* is a negative regulator of the JNK pathway and acts as an inhibitor of apoptosis in insect cells. We initially analyzed the relationships between the published vRNA loads [15] and GOI expression levels and then further characterized the role of *Aae DSH* (*aaeDSH*) using dsRNA gene silencing.

## 2. Materials and Methods

### 2.1. Mosquitoes, Viral Infections and Sample Preparation

The Poza Rica strain of *Aae* was colonized in 2012, originating from the state of Veracruz, Mexico [23]. The *Cxt* Kern National Wildlife Refuge (KNWR) colony [24], established in 1952, was obtained from the Centers for Disease Control and Prevention (Fort Collins, CO, USA). Mosquito colonies were maintained at 24–26 °C (*Cxt*) or 28 °C (*Aae*) on a 12:12 light:dark cycle; adults were fed water and sucrose ad libitum. Larvae were reared on TetraMin fish food (Spectrum Brands Pet, Blacksburg, VA, USA) ground in a coffee grinder.

Archived mosquito samples (3–7 days old) were orally exposed to RVFV MP-12 as described in Campbell et al. [15]. In brief, freshly grown MP-12 RVFV was grown in Vero cells and then mixed 1:1 in defibrinated calf blood (Colorado Serum Company, Denver, CO, USA), with 1 mM ATP and 0.075% sodium bicarbonate as phagostimulants and orally provided to adult mosquitoes in water-jacketed feeders. Archived RVFV MP-12 infected individuals from the previous project were stored at −80 °C in 100 μL mosquito diluent (DMEM, 20% heat-inactivated FBS, 50 μg/mL Pen-Strep, 50 μg/mL gentamicin and 2.5 μg/mL amphotericin B). Viral bloodmeal titers were at 6.4–6.7 log_10_ plaque-forming units (PFU) per ml for the previously published study. As a control, a separate group of age-matched mosquitoes were provided with a non-infectious bloodmeal 3–7 days post-emergence and held for the indicated period. For both *Aae* and *Cxt*, 3 replicates of 20 individual mosquito bodies infected with RVFV MP-12 strain were analyzed and compared to 3 replicates of 20 individual uninfected blood-fed mosquito bodies. For the *DSH* gene silencing experiment (detailed below), virus meals averaged 6.3–7.0 log_10_ PFU/mL. Mosquitoes were held for 3, 7 or 14 days post-challenge (dpc) at 28 °C and 70–80% humidity. 

### 2.2. Host Cell RNA Extraction

RNA was extracted from 100 μL of each (originally 250 μL) individual mosquito carcass homogenate, collected in Campbell et al. [15], following the manufacturer’s recommendations. In brief, 900 μL TRIzol reagent (Invitrogen, Waltham, MA, USA) and 1 μL linear acrylamide (Invitrogen, Waltham, MA, USA) were added to each sample homogenate. Next, 180 μL chloroform was added, and samples were vortexed to mix thoroughly and centrifuged at 12,000× *g* at 4 °C for 15 min. The aqueous phase was added to 1.7 mL conical tubes containing 450 μL isopropyl alcohol and samples were incubated at room temperature for 10 min. After incubation, samples were centrifuged at 12,000× *g* at 4 °C for 10 min. RNA pellets were washed with 80% ethyl alcohol. RNA pellets were resuspended in 25 μL nuclease-free water and then immediately used for reverse transcription or stored at −80 °C.

### 2.3. Reverse Transcription Real-Time Quantitative PCR

Primers, listed in Appendix A, were designed in Geneious (version 2020.2.4) using sequence information from vectorbase.org [25] for *Aae* and sequences from *Cxt*, which were identified by sequence similarity alignments with *Cx quinquefasciatus* [26]. RNA was reverse transcribed using the QuantiTect Reverse Transcription kit (Qiagen, Germantown, MD, USA), following the manufacturer’s protocol and random hexamers. Then, 12 μL RNA was used for reverse transcription; 2 μL of 1:10 diluted cDNA was used for each 20 μL reaction. Quantitation of GOI transcript expression levels was performed via RT-qPCR using gene-specific primers (Appendix A) and the QuantiFast SYBR Green PCR kit (Qiagen, Germantown, MD, USA) and manufacturer’s recommendations. The Quantitect system is designed to quantify a variety of genes across a wide linear range. Nevertheless, it is possible that the use of a single annealing temperature contributed to alterations of the amplification efficiency across targets. RT-qPCR conditions: one cycle 95 °C for 15 min, followed by 40 cycles of denaturation at 94 °C for 15 s, then annealing at 60 °C for 30 s, then 72 °C for 30 s, finished by one cycle of melt curve at 95 °C for 1 s, followed by 60 °C for 20 s then 95 °C for 1 s. All samples were run in duplicate in hard-shell high profile 96-well semi-skirted PCR plates (Bio-Rad Laboratories, Inc., Hercules, CA, USA). All RT-qPCR were carried out on the QuantStudio^TM^ 3-96-Well 0.1-mL Block (Applied Biosystems, Waltham, MA, USA). Expression levels were determined using the delta–delta Ct method, comparing GOI delta Ct levels to those of averaged delta Ct levels of bloodfed controls. *ACTIN* and *RPS7* (ribosomal protein S 7) reference standards were used. In the case of *Cxt*, inconsistent *RPS7* amplification was obtained, so *ACTIN* alone was used as a reference standard. Viral RNA copy numbers were determined as previously reported in Campbell et al. [15].

### 2.4. Gene Silencing of DSH by dsRNA Injection

An approximately 1200 base-pair region of the *DSH* gene was synthesized and cloned into pUC-IDT-AMP Goldengate cloning vector (Integrated DNA Technologies, Coralville, IA, USA). PCR amplification of *DSH* dsDNA fragments containing flanking T7 RNA polymerase promoters (Appendix A) was performed and then gel-purified. Next, dsRNA was transcribed using standard methods [27]. Resulting dsRNA was diluted in phosphate-buffered saline (PBS) to a concentration of 3000 ng/µL. Adult *Aae* females (3–5 days post emergence) were fed an artificial infectious bloodmeal containing a two-fold dilution of the RVFV-MP-12 virus, then held on ice to sort out engorged females and injected simultaneously in the thorax with ~250 ng dsRNA using a Nanoject III (Drummond Scientific, Broomall, PA, USA). For 3, 7 and 14 dpc, 3 replicates of 20 individual RVFV MP-12-exposed *DSH* dsRNA-inoculated mosquito bodies were analyzed and compared to 3 replicates of 20 *eGFP* dsRNA inoculated RVFV MP-12-exposed individuals. In addition, a separate dsRNA–injection experiment was performed to confirm *DSH* dsRNA silencing. Next, qRT-PCR was performed, following the manufacturer’s recommendations (QuantiFast SYBR Green PCR kit (Qiagen, Germantown, MD, USA)). The delta–delta Ct method was used to calculate the reduction in *DSH* expression, using *RPS7* as an internal reference standard and GFP-dsRNA injected mosquitoes as controls [28].

### 2.5. Viral Copy Number Determination

RVFV viral copy numbers were determined via one-step RT-qPCR of the RVFV L segment using TaqMan probes and a standard curve, as previously described [15]. Next, 15 μL of Applied Biosystems TaqMan^®^ Fast Virus 1-Step PCR Master Mix, catalog #4444434 (Applied Biosystems, Waltham, MA, USA), was used with 5 μL of mosquito RNA for each 20 μL of reaction. All samples were run in duplicate in hard-shell high profile 96-well semi-skirted PCR plates (Bio-Rad Laboratories, Inc., Hercules, CA, USA), using the QuantStudio^TM^ 3-96-Well 0.1-mL Block (Applied Biosystems, Waltham, MA, USA). One-step RT-qPCR conditions: one cycle of 50 °C for 5 min, 95 °C for 20 s, followed by 40 cycles of 95 °C for 3 s then annealing/extension step at 60 °C for 30 s. RVFV copy numbers in *DSH* dsRNA-silenced samples were adjusted per nanogram of input RNA. Sample means were compared to those of *eGFP* dsRNA-silenced control samples to determine significant differences using *t*-test (Prism Graphpad, version 8.1.0).

### 2.6. Cis-Regulatory Region Analysis

Genomic regions 1000 nts upstream of the transcriptional start sites were captured from the *Aae* L5 genebuild (*Aedes-aegypti*-LVP_AGWG_TRANSCRIPTS_AaegL5.2.fa) [29] for each of the five GOIs using SAMtools faidx command [30]. Cis-5′ sequences were assessed for the presence of putative regulatory binding sites using the TFBind web tool (https://tfbind.hgc.jp/) (accessed on 2 March 2023) [31], using ACTIN 5′ regulatory regions as a control. Specific transcription factor (TF) binding sites (TFBSs) were compared to identify those present upstream of specific GOIs but not others. Only those TFBSs that were shared among the GOIs but not *ACTIN* are shown in Appendix A.

### 2.7. Statistical Analysis

All statistical analyses were performed in Prism Graphpad (version 8.1.0).

**Figure 1 viruses-15-02140-f001:**
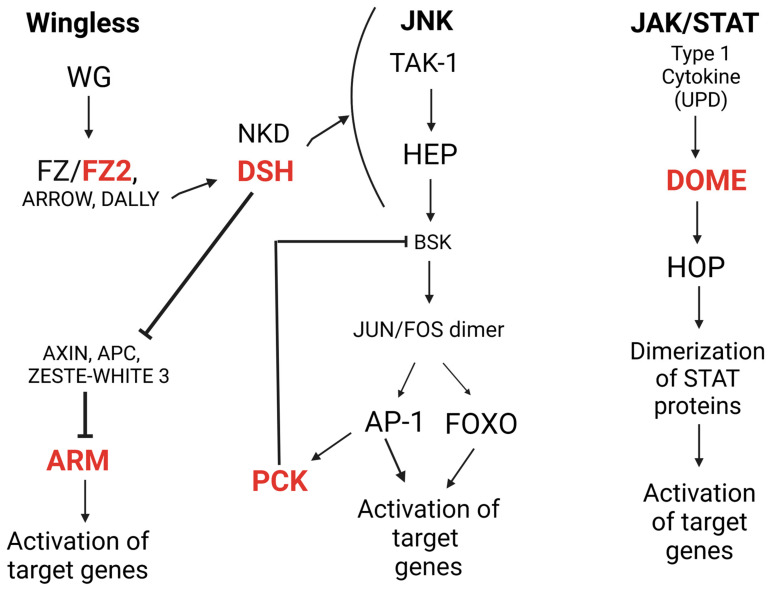
Simplified representation of the cell-signaling pathways interrogated in this project. All three pathways are signal transduction pathways, and as such, mediate transcriptional activation of a variety of downstream effectors or target genes. GOIs in this study are in red font. Dsh mediates crosstalk between the wingless (WNT) and c-Jun N-terminal Kinase (JNK)-signaling pathways [32,33]. Frizzled2 (fz2) acts as a cell surface receptor for Wg, thus initiating signal transduction [34]. Arrow [35] and dally [36], both membrane-bound co-receptors, act in conjunction with fz2 to transmit the Wg signal. Dishevelled (dsh) is a cytosolic protein that is activated upon interaction with fz2 [37]; dsh downregulates zeste-white 3, which forms a multiprotein complex that negatively regulates armadillo (arm) [38]. Upon downregulation of zeste-white 3 and its associated proteins, arm escapes degradation and then accumulates in the nucleus to transcriptionally activate downstream targets [39]. Dsh, under the control of a naked cuticle, can activate or inhibit the WNT pathway and push signaling to the JNK pathway [40]. Puckered (pck) encodes a MAPK phosphatase that acts as a negative regulator of the JNK pathway, repressing the basket protein [41] in dipteran late-embryogenesis. Domeless (dome) is an IL-6-related cytokine receptor that, upon binding of unpaired 1, induces clathrin-dependent endocytosis of the ligand complexes, and initiation of signaling [42]. Upon cytoplasmic signal transduction, hopscotch auto-phosphorylates, along with the cytoplasmic dome receptor, which creates binding sites for STAT proteins, ultimately leading to the activation of downstream target genes [43,44]. Generated in Biorender.

## 3. Results

### 3.1. Signaling Pathway Transcripts Are Substantially Depleted in Mosquitoes Infected with RVFV MP-12

Differential gene expression analysis of infected *Aae* indicated that *DSH*, *ARM*, *PCK* and *DOME* transcripts were depleted, relative to bloodfed controls, at 14 dpc (Figure 2A). Individual mosquitoes, rather than pools, were assessed, resulting in variability in transcript levels across all GOIs, which is common for mosquito studies [45]. *Cxt* showed similar profiles, with depletion of *DSH*, *FZ2*, *PCK* and *DOME* (Figure 2B).

### 3.2. DSH Expression Changes Are Associated with Viral Loads

We then compared vRNA-positive or negative individuals to those relative gene expression values reported in Figure 2A. *Aae* showed significant differences in *DSH* transcript levels relative to the presence or absence of vRNA (Figure 3A). No other transcripts showed a correlation between the presence of vRNA and GOI expression patterns. Further analysis showed an inverse correlation between vRNA levels and *aaeDSH* expression pattern for carcasses but not saliva samples from the same mosquitoes (Appendix A).

### 3.3. Silencing of DSH Increases MP-12 vRNA Copy Numbers

To further characterize the role of *aaeDSH* during RVFV infection, we performed gene silencing using dsRNA injection of *Aae* adults and then measured the vRNA copy number at intervals post-challenge. The silencing of *DSH* was confirmed by RT-qPCR of individual mosquitoes at 3 days post-injection (Appendix A, mean 99% reduction compared to controls). We saw nearly a one log increase in viral genome copies in *DSH*-silenced mosquitoes compared to GFP-injected controls at 3 dpc (*t*-test, *p* < 0.05, Figure 3B); this effect waned by 7 dpc.

### 3.4. Analysis of Cis-Regulatory Regions in Aae

The observation of sustained transcript depletion for GOIs and the inverse correlation of *aaeDSH* to vRNA genome copies led us to question whether the assessment of genomic *cis*-regulatory regions might provide insight into the underpinnings of the transcript repression reported above. Therefore, for *Aae*, we interrogated the 1000 nucleotides upstream of each GOI [29] for the presence of *cis*-regulatory regions that were not present in the 5′ region of the control gene *ACTIN* (Appendix A) [31]. Our analysis revealed 12 TFBSs of interest. Of these, three have been validated as direct or epistatic targets of the STAT transcription factor. Secondly, all five GOIs contained sterol regulatory element-binding protein 1 (SREBP) TFBSs, whereas *ACTIN* did not. In addition, the TFBS (HEN1) for HEN1 2′ o-methyltransferase was found in the upstream region of *ARM*, *PCK* and *FZ2* but not *DSH*. We were unable to assess *Cxt* upstream cis-genomic regions due to the provisional nature of the *Cxt* genome.

## 4. Discussion

Here, we describe GOI differential transcript expression patterns for key effectors of *Aae* (low competence) and *Cxt* (high competence) viral immune response and signaling pathways at 14 dpc following RVFV MP-12 exposure. Notably, all significantly differentially expressed GOIs were downregulated compared to uninfected controls, however, specific expression patterns differed between mosquito species. Components of the WNT pathway, *FZ2* and *DSH*, were substantially reduced in *Cxt* at 14 dpc, whereas *ARM* and *DSH* were reduced in *Aae* (Figure 2). Of note, *ARM*, the beta-catenin ortholog, was not significantly depleted in *Cxt*. *ARM* has been implicated as a direct target of RVFV transcriptional activation [16]. So, it is tempting to speculate that its lack of depletion in *Cxt* is directly due to viral activity that may affect vector competence phenotypes [15].

Our findings, combined with evidence from earlier reports, are consistent with the hypothesis that *aaeDSH* has nuanced regulatory roles that are not dependent on *FZ2* or *FZ*. An inverse correlation between viral copies and *aaeDSH* levels was identified (Appendix A), indicating that higher levels of *aaeDSH* are associated with reduced viral loads. Interestingly, no association was found between viral copy numbers and expression of any GOI in *Cxt*. In addition, silencing of *aaeDSH* resulted in a significant increase in vRNA genome copies (Figure 3B), consistent with a role in antiviral immunity in *Aae*. The effects at 3 dpc were not seen at subsequent timepoints, consistent with waning effects of the dsRNA injection or a feedback loop that stimulated gene expression. In drosophilids, dsh localizes variously to plasma membranes and/or the WNT signalosome [46], where it stimulates cell signaling that activates cell proliferation, self-regeneration or development, depending on specific cell-signaling stimuli [46,47]. Though fz2 and dsh physically interact in vivo as part of WNT signaling, our results showed species-specific effects of gene expression levels following RVFV exposure. Displacement of dsh from the WNT/axin signalosome promotes signalosome to degradasome conversion [48]. In mammals, naked cuticle interacts with dsh to shift its activity away from WNT signaling toward stimulation of the JNK immune pathway [40]. Of note, JNK activity has been shown to restrict dengue, Zika and chikungunya infection in *Aedes* salivary glands [21]. Though the mechanisms of naked cuticle and dsh interactions have not been explored in mosquitoes, our findings provide the first hint that this could be a fruitful area for further exploration. Our study suggests that regulation of *aaeDSH* may be one chokepoint for viral susceptibility/refractorinesss, though further work needs to be done to define exactly how dsh activity mediates vector competence phenotypes.

We were particularly interested in the differential presence of specific TFBSs upstream of the GOIs as a way to gain clues about their regulation. Therefore, we assessed genomic regions 1000 bp upstream of *Aae* GOIs. All five GOIs contained SREBP-binding sites. SREBP1 regulates genes responsible for lipid and cholesterol production [49]. Fluctuations in lipid metabolism occur during arbovirus infection of mosquitoes (reviewed in [50]). Specifically, in *Aae*, fatty acid synthase 1 (*FAS1*) facilitated DENV2 replication in mosquito midguts [51]. Fas1 is a multifunctional enzyme that produces fatty acids for membrane assembly and repair [52]. More work is needed to determine whether or not SREBP1 or other TFBS sites may be regulated by an unidentified viral protein to alter lipid production, favor viral replication and repress GOI expression.

Fz2 and dsh proteins physically interact in vivo during WNT signaling [47], however, our data showed that pathway component regulation differs in a species-specific manner. For example, *FZ2* was not depleted in *Aae*. We also saw significant decreases for both *Cxt* and *Aae* in expression of the proviral transcript *PCK*, which is a regulator of the anti-viral JNK pathway [41]. However, by testing whole mosquitoes (sans saliva and legs/wings), we may have missed compartment-specific regulatory differences. For example, Chowdhury et al. reported that *PCK* expression was enriched in salivary glands at 7 days following exposure to chikungunya, dengue 2 or Zika viruses, with no significant change at 14 dpc [21]. It is not currently known whether suppressed transcript levels, e.g., *DSH* and proviral effectors, e.g., *PCK*, were due to efforts by the mosquito to control viral amplification or due to viral hijacking of host gene expression. 

These results, and those from Harmon et al. [16], support the idea that RVFV infection is altered by WNT signaling in both humans and mosquitoes, however, with different effects. In *Cxt*, the more competent mosquito vector, cxt*DSH* transcript levels, though suppressed compared to uninfected controls, were not correlated with the viral genome copy number, in contrast to *Aae*. The less competent vector, *Aae*, showed a negative correlation between viral infection and *aaeDSH* expression. In contrast, *cxtARM*, in the more competent strain, was not depleted. Moreover, Harmon et al. reported induction of WNT signaling at beta-catenin (*ARM* ortholog) during RVFV infection of human cell cultures [16]. It is possible that lack of *cxtARM* depletion is analogous to beta-catenin activation that occurs in humans. It is currently unclear why this pathway is differentially triggered in the arthropod vector compared with a vertebrate host, which also warrants further exploration.

## 5. Conclusions

In this study, expression of five GOIs (dsh, fz2, arm, pck and dome) were significantly decreased following challenge of *Aae* or *Cxt* with RVFV MP-12, suggesting RVFV infection leads to prolonged changes in host cell-signaling response pathways. We speculate that differences in gene expression regulatory patterns may be related to overall differences in virus susceptibility between these two species. Initial analysis of cis-regulatory regions suggests that further analysis of *Aae* and *Cxt* hosts should be done to gain a better understanding of the transcriptional regulatory processes that underpin vector competence.

## Figures and Tables

**Figure 2 viruses-15-02140-f002:**
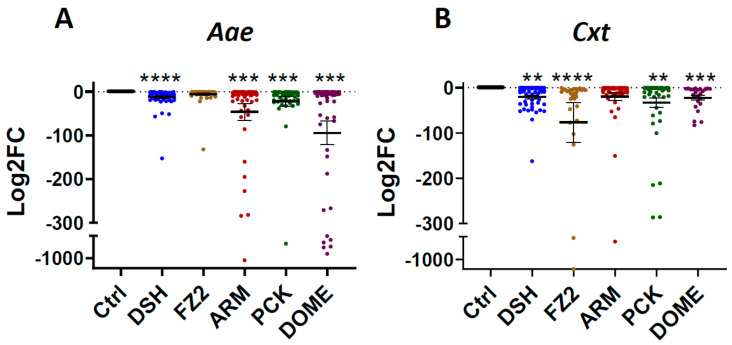
Selected viral response genes were significantly downregulated in both (**A**) *Aae* and (**B**) *Cxt* RVFV MP-12-infected mosquito carcasses, relative to uninfected controls at 14 dpc. Mosquitoes were exposed to RVFV MP-12 and held for 14 days prior to collection of bodies for RNA isolation and RT-qPCR. Each symbol represents an individual mosquito. Significance determined by *t*-test (** *p* < 0.01, *** *p* < 0.001, **** *p* < 0.0001) of the delta Ct values of the experimental and bloodfed control groups. Data represent 3 biological replicates of 20 samples each. Numbers of samples in the graph are as follows, Aae bloodfed *n* = 60, *aaeDSH* = 60, *aaeFZ* = 47, *aaeARM* = 60, *aaePCK* = 50, *aaeDOME* = 51; Cxt bloodfed *n* = 60, cxt*DSH* = 52, *aaeFZ* = 30, *aaeARM* = 52, *aaePCK* = 48, *aaeDOME* = 24. Bars indicate standard error of the mean. Relative gene expression levels (Log2FC) were calculated using the delta–delta Ct method by normalizing expression to housekeeping gene(s) (See Section 2).

**Figure 3 viruses-15-02140-f003:**
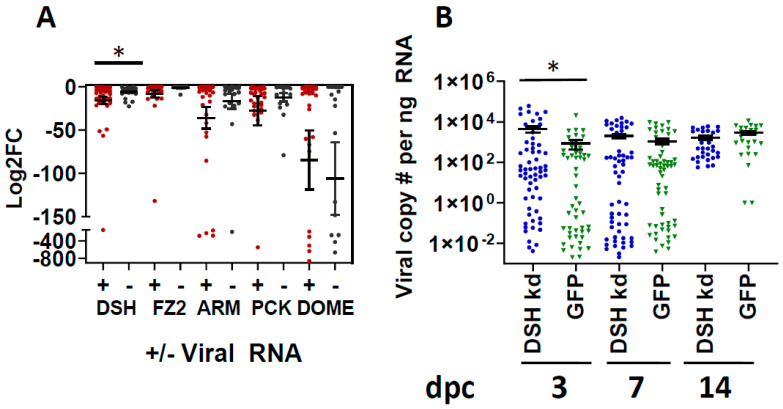
*Aae DSH* expression is associated with vRNA copy number. (**A**) Log2 fold-change (Log2FC) data from Figure 2A were replotted after grouping mosquitoes by viral positivity using previously published viral copy number data [15]. Each symbol represents an individual mosquito. Significance determined by *t*-test (*, *p* < 0.05) of the delta Ct values of the experimental and bloodfed control groups. Bars indicate standard error of the mean. Data represent 3 biological replicates. Sample numbers are the same as shown in Figure 2A. A Ct threshold cut-off of 38 was used for determination of vRNA positivity. (**B**) Silencing *DSH* increases viral loads in *Aae*. *Aae* mosquitoes were fed a bloodmeal containing RVFV MP-12. Engorged females were injected with either *DSH* or *GFP* dsRNA during post-bloodmeal sorting and held for 3, 7 or 14 days prior to collection for RNA extraction and qPCR of viral L segment copy numbers. Viral genome copies were significantly higher in *DSH*-silenced individuals than controls (*t*-test, *, *p* < 0.05). A total of 3 independent experimental replicates of 20 mosquitoes each were used. Numbers of samples in the graph are as follows, 3 dpc- *aaeDSH* dsRNA = 60, *GFP* dsRNA = 51; 7 dpc-*aaeDSH* dsRNA = 58, *GFP* dsRNA = 60; 14 dpc were one replicate of the following- *aaeDSH* dsRNA = 38, *GFP* dsRNA = 28. Error bars indicate mean viral copy numbers with standard error of the mean.

## Data Availability

Raw data available upon request.

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
