# Peer review of "Dishevelled Has Anti-Viral Activity in Rift Valley Fever Virus Infected Aedes aegypti"

_viruses, 2023, doi:10.3390/v15112140_

Round 1

Reviewer 1 Report

In this study the authors examined whether the expression of several genes in the Wnt/beta-catenin, JAK-STAT, and JNK signaling pathways was different in RVFV-infected Aedes and Culex mosquitoes compared to blood fed controls, using archived mosquito samples from an earlier study. They found that several genes were significantly downregulated in both mosquito species upon infection at 14 dpi.  Comparison with virus RNA levels indicated that only DSH downregulation was correlated with the presence of viral RNA, and only in Aedes.  DSH expression also correlated with the level of viral RNA in bodies but not in saliva.  The authors then used RNAi to silence DSH expression in Aedes mosquitoes that had fed on blood containing RVFV.  A small but statistically significant increase in mean viral RNA levels was observed in the mosquitoes that received DSH dsRNA compared to control.  This increase was seen at 7 dpi but not at 14 dpi.  The authors then examined the promoter regions of the genes that were surveyed and observed some potential transcription factor binding sites of interest.

Overall this is a fairly straightforward and logical study that provides an advancement in our knowledge of mosquito immunity against arboviruses.  I have some comments that can improve the manuscript.

1.  Line 33, The words fever and virus should not be capitalized in the virus name Rift Valley fever virus.  Only proper nouns should be capitalized in virus names.  See the ICTV recommendations for writing virus names at https://ictv.global/faqs.

2. There are issues with some of the references in the manuscript.  References 17 and 18 do not appear related to the statement on line 58.  Ref 17 does not mention the Wnt pathway, and ref 18 does not mention viruses.  Also, Ref 19 is not related to the statement on line 71.  I did not check all of the references but the authors should carefully check them to make sure they are referring to the correct paper.

3. Line 194, instead of saying genes are depleted, I think the authors meant to say that gene transcripts are depleted.

4. I appreciated the fact that the authors measured gene expression in individual mosquitoes; too many studies use pooled material which can be misleading.  However, I would like to see the gene silencing data, which is not provided.  The authors stated that there was a mean 93% reduction, but it would be useful to know how variable the gene silencing was, in order to properly evaluate the results of the experiment.  This could be provided either as an additional figure panel, a table or a supplemental figure or table.

5. An increase in virus genome copies of greater than 2-fold is questionable when it comes to its biological significance.  Just because a result has a p value less than 0.5 does not necessarily mean it is biologically meaningful. However, just looking at the graph in Fig 4, it appears that the authors are perhaps being conservative in their estimate.  It appears that there is closer to a 10-fold difference in the mean titers at 7 dpi.  Is this correct?

7. I have a minor quibble about using the term dpi when referring to mosquitoes that were infected by blood meal.  I prefer to use days post blood meal, since we don’t really know for sure exactly when the infection started in each mosquito, we only know when they took the blood meal.

Author Response

We appreciate the Reviewer’s helpful comments and suggestions. Please find our responses below.

  1. Line 33, The words fever and virus should not be capitalized in the virus name Rift Valley fever virus.  Only proper nouns should be capitalized in virus names.  See the ICTV recommendations for writing virus names at https://ictv.global/faqs.

This correction has been made.

  1. There are issues with some of the references in the manuscript.  References 17 and 18 do not appear related to the statement on line 58.  Ref 17 does not mention the Wnt pathway, and ref 18 does not mention viruses.  Also, Ref 19 is not related to the statement on line 71.  I did not check all of the references but the authors should carefully check them to make sure they are referring to the correct paper.

Thank you for catching the formatting glitch. All references have been carefully checked, confirmed and corrected as needed.

  1. Line 194, instead of saying genes are depleted, I think the authors meant to say that gene transcripts are depleted.

This correction has been made.

  1. I appreciated the fact that the authors measured gene expression in individual mosquitoes; too many studies use pooled material which can be misleading.  However, I would like to see the gene silencing data, which is not provided.  The authors stated that there was a mean 93% reduction, but it would be useful to know how variable the gene silencing was, in order to properly evaluate the results of the experiment.  This could be provided either as an additional figure panel, a table or a supplemental figure or table.

This correction has been made- see new Supplemental File 1. Unfortunately, use of individual mosquitoes complicated our assessment of the effects of DSH transcript silencing on vRNA loads. See below.

  1. An increase in virus genome copies of greater than 2-fold is questionable when it comes to its biological significance.  Just because a result has a p value less than 0.5 does not necessarily mean it is biologically meaningful. However, just looking at the graph in Fig 4, it appears that the authors are perhaps being conservative in their estimate.  It appears that there is closer to a 10-fold difference in the mean titers at 7 dpi.  Is this correct?

Our apologies for the confusing representation of the data. Also, thank you for pointing out the need for more investigation. The original graph showed median values, which did not match the statistical test performed. Therefore, we re-graphed the values using means with SEM error bars. At the same time, we discovered a mathematical error in one of the formulas for calculating vRNA copy numbers that has now been corrected. As a result, the 7d timepoint fold-change for DSH- dsRNA-treated samples vs controls is no longer statistically significant, though, it remained around a 2-fold increase, as originally reported. To address your reasonable concern about fold-change, we went back and repeated the experiment to include the 3 day-post-challenge data. The 3 day timepoint showed a more dramatic change in DSH-silenced vs controls. See the new Fig 3b.

  1. I have a minor quibble about using the term dpi when referring to mosquitoes that were infected by blood meal.  I prefer to use days post blood meal, since we don’t really know for sure exactly when the infection started in each mosquito, we only know when they took the blood meal.

We have altered the verbiage to indicate “days post-challenge" to address your concern.

Reviewer 2 Report

The manuscript by Smith et al. evaluated the differential expression of five genes of interest (ARMFZ2DSHDOME, and PCK) in relationship to RVFV RNA loads 14 days after oral infection with the M12 strain. In addition, DSH silencing in Aedes aegypti was used to evaluate if RVFV-responsive host cell signaling processes contribute to variability in vector competence. I recommend this manuscript for publication after revision. The section below details some specific line-by-line comments.

General comments:

1.     Follow the correct gene nomenclature throughout the manuscript:

Do not use Hyphens in gene symbols (e.g., INT1, not TNT-1), except for C. elegans gene symbols. Italicize gene symbols (e.g., ARM), Genotypes, mRNAs, or cDNAs but do NOT italicize gene names (e.g., “armadillo”) or Phenotypes.Capitalization of gene and protein symbols should be styled according to species. See official NCBI Gene full names and symbols for examples (https://www.ncbi.nlm.nih.gov/gene/). Usually, full names of genes and proteins start with a lowercase letter unless they begin with a person’s name (describing a disease/phenotype) or a capitalized abbreviation.

2.     Italicize Latin terms (e.g., ad libitum (Line 84))

3.     For all figures, please indicate the sample size (n=) used in the graphs and statistical analyses. 

Line-by-line comments:

Lines 40-41: It would be ideal to include citations showing a wider range of mosquito species implicated as RVFV vectors among additional geographical locations. Authors can assess these recent publications as an example https://doi.org/10.3390/pathogens11050503https://doi.org/10.1002/vms3.941

Lines 54-55: Please elaborate if the WNT/INT1 viral mediated response occurs only among Bunyaviruses or if this is a widespread phenomenon across several viral orders. In addition, please briefly describe how (identified or hypothesized mechanism) viruses manipulate the WNT/beta-catenin pathways to promote infection. 

Lines 62-67: Although shown in Figure 1, the connection between JNK and the other GOIs is unclear in the introduction. Please briefly mention how the JNK crosstalk with the WNT/beta-catenin pathway. Also, is the mosquito JNK involved in the antiviral response to RVFV infection?

Lines 85-94: Please clarify if the control and RVF MP-12 orally females were aged matched. Or if both groups were fed at different times post-emergence?

Lines 94-97: Please clarify if the archive samples are infected mosquitoes or cell cultures. If mosquitoes, briefly explain how they were infected. 

Lines 121-126: It is surprising to see just one constant annealing temperature of 60℃ for all primer sets across both mosquito species. Is there a potential variation in reaction efficiencies? If so, this needs to be discussed and accounted for in the following expression analyses. 

Lines 146-147: A dsRNA injection of what? Blood-fed controls? Or aged-matched non-fed controls?

Lines 175-192: please mention what is referred to as “target genes” in Figure 1. Not sure if the authors are still referring to GOIs or something else. 

Line 194-209: were all the expression levels of DHSFZ2ARMPCK, and DOME of RVFV MP12-fed mosquitoes included in these analyses? Or were specific values included/excluded depending on the infection status? Or were the mosquitoes that tested positive for RVFV analyzed separately from the ones that tested negative for RVFV at 14dpi? 

Line 212: Since Figure 1 is a schematic representation, I believe the authors refer to Figure 2 instead. 

Lines 215-217: Although a significant p-value was obtained, the R2=0.08 indicates that only 8% of the variance in DSH expression can be explained by variation in viral titers. Please revise the statistical analysis used for the regression analysis (see specific comments linked to Figures 3B and 3C).

Line 225-233: In Figure 3A, please specify the threshold Ct used to determine the positivity of RVFV-fed mosquitoes. In addition, the authors should reanalyze the data in Figures 3B and 3C using a more appropriate model. There is no apparent indication of linearity when observing the distribution of the data and the bad fit obtained for a linear regression (indicated by the R2). If a linearity test was performed, please indicate why a linear regression is appropriate for this data. The authors might consider using logistic regression or a nonlinear regression that generates a curve closest to the data. Also, the expression data were transformed to a log scale, which can further distort the experimental error when using a linear scale. 

Line 281-283: It is also important to mention if other phenotypic effects were observed in mosquitoes after DSHsilencing, for example, flight patterns or wing movements. In flies, DSH mutations cause the improper orientation of body and wing hairs and altered social behaviors. 

Author Response

 We appreciate the Reviewer 2’s helpful comments and suggestions. Please find our responses below.

  1. Follow the correct gene nomenclature throughout the manuscript:

Do not use Hyphens in gene symbols (e.g., INT1, not TNT-1), except for C. elegans gene symbols. Italicize gene symbols (e.g., ARM), Genotypes, mRNAs, or cDNAs but do NOT italicize gene names (e.g., “armadillo”) or Phenotypes.Capitalization of gene and protein symbols should be styled according to species. See official NCBI Gene full names and symbols for examples (https://www.ncbi.nlm.nih.gov/gene/). Usually, full names of genes and proteins start with a lowercase letter unless they begin with a person’s name (describing a disease/phenotype) or a capitalized abbreviation.

These corrections were made.

  1. Italicize Latin terms (e.g., ad libitum(Line 84))

 These corrections were made.

  1. For all figures, please indicate the sample size (n=) used in the graphs and statistical analyses.

 These details are now included.

Line-by-line comments:

Lines 40-41: It would be ideal to include citations showing a wider range of mosquito species implicated as RVFV vectors among additional geographical locations. Authors can assess these recent publications as an example https://doi.org/10.3390/pathogens11050503, https://doi.org/10.1002/vms3.941. 

 Thank you for pointing that out. We added a few sentences to the Introduction to expand the number of mosquito species.

Lines 54-55: Please elaborate if the WNT/INT1 viral mediated response occurs only among Bunyaviruses or if this is a widespread phenomenon across several viral orders. In addition, please briefly describe how (identified or hypothesized mechanism) viruses manipulate the WNT/beta-catenin pathways to promote infection. 

We’ve added a brief statement to this section to emphasize the wide variety of viruses with association to WNT pathway activation, best studied in mammals. We already had a section in the Discussion describing the transcriptional activation of beta-catenin by RVFV, but we have added it to the Introduction, as well, to emphasize the connection. However, we did not go into detail about specific mechanisms of other virus activation of WNT to avoid unnecessary distractions from the topic at hand.

Lines 62-67: Although shown in Figure 1, the connection between JNK and the other GOIs is unclear in the introduction. Please briefly mention how the JNK crosstalk with the WNT/beta-catenin pathway. Also, is the mosquito JNK involved in the antiviral response to RVFV infection?

We added a statement to the Intro describing the evidence for JNK involvement in anti-viral immunity in mosquitoes infected with dengue or chikungunya. In the Discussion, we describe in detail the evidence for cross-talk between the JNK and WNT pathways and feel that it is better placed in that section. See lines ~286-294.

Lines 85-94: Please clarify if the control and RVF MP-12 orally females were aged matched. Or if both groups were fed at different times post-emergence?

We’ve added details to the Methods section to clarify the point that mosquitoes were age-matched.

Lines 94-97: Please clarify if the archive samples are infected mosquitoes or cell cultures. If mosquitoes, briefly explain how they were infected. 

All archived samples were mosquitoes that had been challenged with oral viral meals. We’ve altered the text to clarify this point. “Archived mosquito samples (3-7 days old) were orally exposed to RVFV MP-12 as described in Campbell et al.”

Lines 121-126: It is surprising to see just one constant annealing temperature of 60℃ for all primer sets across both mosquito species. Is there a potential variation in reaction efficiencies? If so, this needs to be discussed and accounted for in the following expression analyses. 

The Qiagen Quantitect RT-qPCR reagent system is specifically designed to quantify a variety of transcripts across a wide linear range and recommends the use of the 60C annealing temperature. Nevertheless, due to the formal possibility that some amplification efficiency issues might be of concern, we added a statement to this effect at ~ line 135.

Lines 146-147: A dsRNA injection of what? Blood-fed controls? Or aged-matched non-fed controls?

We state in this section that the comparison was RVFV MP-12 infected DSH dsRNA-inoculated mosquitoes versus RVFV MP-12 infected eGFP dsRNA-inoculated. We made a small change to clarify the point.

Lines 175-192: please mention what is referred to as “target genes” in Figure 1. Not sure if the authors are still referring to GOIs or something else. 

GOIs are indicated in red in the figure. We’ve added a statement to the legend to indicate that target genes are a variety of downstream effector genes to clarify this point and includes genes that are not the GOIs.

Line 194-209: were all the expression levels of DHSFZ2ARMPCK, and DOME of RVFV MP12-fed mosquitoes included in these analyses? Or were specific values included/excluded depending on the infection status? Or were the mosquitoes that tested positive for RVFV analyzed separately from the ones that tested negative for RVFV at 14dpi? 

Fig 2 reports all GOI expression levels, regardless of the presence of RVFV. Fig 3A shows Aae GOI expression levels, grouped by the presence/absence of RVFV vRNA (L segment). Cxt showed vRNA in all samples, so they were not appropriate for this assessment.

Line 212: Since Figure 1 is a schematic representation, I believe the authors refer to Figure 2 instead. 

                Thank you for pointing out the error. The correction was made.

Lines 215-217: Although a significant p-value was obtained, the R2=0.08 indicates that only 8% of the variance in DSH expression can be explained by variation in viral titers. Please revise the statistical analysis used for the regression analysis (see specific comments linked to Figures 3B and 3C).

After consulting with a statistician, the analysis was repeated using Spearman rank correlation, which is a non-parametric method and should be more informative than the linear regression originally used. The Spearman correlation results were also statistically significant. We also moved this figure to the Supplemental Figure. Using this analytical method, the values were n=60, R = -0.293, p=0.0231, which is an improvement over the previous significance.

Line 225-233: In Figure 3A, please specify the threshold Ct used to determine the positivity of RVFV-fed mosquitoes.

Done.

 In addition, the authors should reanalyze the data in Figures 3B and 3C using a more appropriate model. There is no apparent indication of linearity when observing the distribution of the data and the bad fit obtained for a linear regression (indicated by the R2). If a linearity test was performed, please indicate why a linear regression is appropriate for this data. The authors might consider using logistic regression or a nonlinear regression that generates a curve closest to the data. Also, the expression data were transformed to a log scale, which can further distort the experimental error when using a linear scale. 

Please see our response above for Ln 215-217. Unfortunately, the requested logistic regression is inappropriate because it is designed for categorical variables.

Line 281-283: It is also important to mention if other phenotypic effects were observed in mosquitoes after DSHsilencing, for example, flight patterns or wing movements. In flies, DSH mutations cause the improper orientation of body and wing hairs and altered social behaviors. 

It’s true that DSH silencing in Drosophila causes developmental deficiencies, however we silenced the gene in adults. Though not specifically tested for- no mortality or overt behavioral differences were observed.

Round 2

Reviewer 1 Report

The authors have adequately addressed my comments for the most part, however I am a bit confused by the new supplemental Fig. 1A.  The bar graph appears to indicate a mean 99% silencing, but the authors state it as being 93% in the text. Also, please provide the n, R and p values for the saliva results in Supp Fig 1C in the legend (as was done for the bodies).

Author Response

The authors have adequately addressed my comments for the most part, however I am a bit confused by the new supplemental Fig. 1A.  The bar graph appears to indicate a mean 99% silencing, but the authors state it as being 93% in the text. Also, please provide the n, R and p values for the saliva results in Supp Fig 1C in the legend (as was done for the bodies).

Thank you for your feedback. The requested changes have been made to the Supplemental Figure legends at the end of the manuscript. Specifically, the percent reduction in dsh silencing has been corrected to 99%, as indicated in the graph, and the statistical values have been added to the saliva results.